# Causal Mediation Analysis with Multi-dimensional and Indirectly Observed Mediators

## Abstract

Causal mediation analysis (CMA) is a powerful method to dissect the total effect of a treatment into direct and mediated effects within the potential outcome framework. This is important in many scientific applications to identify the underlying mechanisms of a treatment effect. However, in many scientific applications the mediator is unobserved, but there may exist related measurements. For example, we may want to identify how changes in brain activity or structure mediate an antidepressant's effect on behavior, but we may only have access to electrophysiological or imaging brain measurements. To date, most CMA methods assume that the mediator is one-dimensional and observable, which oversimplifies such real-world scenarios. To overcome this limitation, we introduce a CMA framework that can handle complex and indirectly observed mediators based on the identifiable variational autoencoder (iVAE) architecture. We prove that the true joint distribution over observed and latent variables is identifiable with the proposed method. Additionally, our framework captures a disentangled representation of the indirectly observed mediator and yields an accurate estimation of the direct and mediated effects in synthetic and semi-synthetic experiments, providing evidence of its potential utility in real-world applications.

## 1 Introduction

Causal inference methods are powerful tools to understand and quantify the causal relationships between treatments and outcomes, motivating studies in many areas (Athey & Imbens, 2017; Glass et al., 2013; Imai et al., 2010; Rothman & Greenland, 2005). Causal inference has been combined with machine learning in recent years to make powerful and flexible frameworks (Guo et al., 2020; Li & Zhu, 2022). While these frameworks are highly useful to estimate the total treatment effect on an outcome, many scientific applications require understanding *how* a treatment impacts outcomes. This knowledge can then be used to design interventions that target intermediate variables to influence the outcome of interest. For example, we may want to identify neural changes that mediate a behavioral outcome when studying a treatment for a psychiatric disorder. Recent work has in fact found and *manipulated* neural changes related to depression (Hultman et al., 2018) and social processing (Mague et al., 2022).

This need motivates the usage of *causal mediation analysis* (CMA), which estimates the causal effect on an outcome of interest that is due to changes in intermediate variables (the "mediators") versus directly from the treatment (Pearl, 2001). In specific contexts, understanding the role of the mediator is crucial as it tells us how nature works and provides insights into the underlying mechanisms that link variables, which enables a more accurate assessment of the treatment's effectiveness. In the above case, this means estimating how much of the behavior change is explained by the treatment's impact on the brain, as well as how much behavioral change is unexplained by that relationship. Early studies on mediation analysis mainly adopted linear structural equation models (SEMs) including Wright's method of path analysis (Wright, 1923; 1934) and Baron and Kenny's method for testing mediation hypotheses (Baron & Kenny, 1986). In the past few decades, researchers have come up with nonparametric generalizations for SEMs (Balke & Pearl, 2013; Jöreskog et al., 1996) which do not impose any functional or distributional forms on the causal relationships and therefore offer greater flexibility in modeling complex dependencies between variables.

Despite these advances, a key challenge is that causal mediation analysis typically assumes a low-dimensional, often one-dimensional, mediator, whereas in many cases we want to identify mediation effects of complex data, such as neuroimaging, electrophysiology, and myriad -omics studies. In this paper, we build upon the concept of the identifiable variational autoencoder (iVAE) (Khemakhem et al., 2020) and introduce a novel framework for CMA that can handle *multi-dimensional* and *indirectly observed* mediators. We assume that there is a latent space that generates the high-dimensional observed data (e.g., a smaller latent space can generate the observed neural dynamics). By using an identifiable model structure, we show that we can recover the latent space prior conditioned on the treatment and any available covariates. In summary, our main contributions are:

- We propose a causal graph that involves both an *indirectly observed* mediator and observed covariates that acts as a confounder for the treatment, the mediator, and the outcome.

- We build a framework for CMA that can handle *multi-dimensional* and *indirectly observed* mediators based on the proposed causal graph.

- We theoretically prove that the joint distribution over observed and latent variables in our framework is identifiable.

- We show that our framework learns a disentangled representation of the *indirectly observed* mediator between control and treatment groups.

- We empirically demonstrate the effectiveness of our framework on complex synthetic and semi-synthetic datasets.

## 2 Related Work

**Causal Mediation Analysis**  As mentioned in the introduction, traditional mediation analysis was mainly based on linear SEMs where the direct, mediated, and total effects are determined by linear regression coefficients (Baron & Kenny, 1986; MacKinnon, 2012; MacKinnon & Dwyer, 1993; Wright, 1923; 1934). Despite its simplicity, this approach relies on several assumptions such as normally distributed residuals (Pearl, 2014) and often leads to ambiguities when either the mediator or the outcome variable is not continuous (Rijnhart et al., 2019). To address this limitation, researchers formulated the causal mediation analysis (CMA) framework based on counterfactual thinking (Holland, 1988; Pearl, 2001; Rubin, 1974), which can accommodate nonlinear or nonparametric models such as targeted maximum likelihood estimation (Zheng & van der Laan, 2012), inverse propensity weighting (IPW) (Huber et al., 2013), and natural effect models (NEMs) (Lange et al., 2012). Within the counterfactual framework, the causal effects are calculated as the difference between two counterfactual combinations of mediators and outcomes, for which we will provide formal definitions in Section 3. Researchers also came up with other metrics for quantifying causal effects such as the natural direct effect among the untreated (Lendle et al., 2013), the population intervention effect (Hubbard & Van der Laan, 2008), and the population intervention indirect effect (Fulcher et al., 2020), but here we stick with Pearl's definition (Pearl, 2001). Although causal effects are defined at the individual level, in practice, we usually relax our estimation to their expected values over the population as we do not generally observe both potential outcomes simultaneously (Holland, 1986).

**Causal Mediation Effect Estimation with Deep Models**  Deep learning models have gained increasing attention for their capability in estimating causal effects within the potential outcome framework (Alaa & Van Der Schaar, 2017; Jiang et al., 2023; Louizos et al., 2017; Shalit et al., 2017). In contrast, the use of deep learning models for mediation effect estimation has received comparatively less exploration. Xu et al. (2022) developed a semiparametric neural network-based framework to reduce the bias in CMA. Cheng et al. (2022) and Xu et al. (2023) used variational autoencoders (VAEs) to estimate the mediation effect based on a causal graph with hidden confounders. Although these VAE-based methods share some similarities with our proposed method, we distinguish ourselves by modeling the *mediator* as the latent variable rather than the covariates, resulting in a different causal graph. Furthermore, these approaches assume that the mediator is observable and one-dimensional, which is not necessarily the case in many scientific applications.

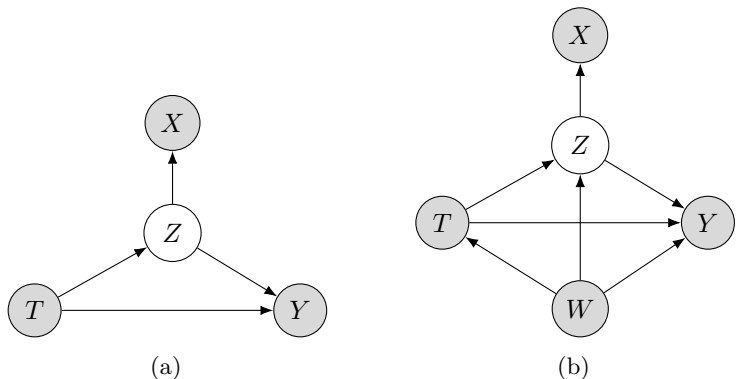

Figure 1: Graphs of CMA for (a) case without observed covariates and (b) case with observed covariates, where $T$ is the treatment assignment, $Y$ is the outcome, $Z$ is the unobserved true mediator, $W$ is a set of observed covariates, and $X$ is a feature caused by the unobserved true mediator $Z$ with a much higher dimension. The observed variables are colored in grey.

**Multi-dimensional Mediators**   Compared to the many CMA methods proposed, significantly less research has been conducted on scenarios where the mediator is multi-dimensional and not directly observable. The majority of investigations on this subject are situated within the domains of neuroscience (Chén et al., 2018; Nath et al., 2023), biostatistics (Zhang et al., 2021a), and bioinformatics (Perera et al., 2022; Yang et al., 2021; Zhang et al., 2021b; 2016). The approach proposed by Nath et al. (2023) is the most relevant work to our research, where the high-dimensional mediator is first transformed into a one-dimensional variable, and the mediation effect is estimated using an iterative maximization algorithm. Nevertheless, all these methods primarily rely on linear SEMs and neglect the impact of any confounding variables, thereby limiting their applicability.

## 3   Problem Setup

We assume that our causal model belongs to one of the two cases as displayed in Figure 1. To ensure consistency with previous studies on mediation analysis (Pearl, 2014; Hicks & Tingley, 2011; MacKinnon et al., 2007), we further assume that the treatment assignment $T$ is binary for each observed samples, with $T = 0$ indicating an assignment to the control group and $T = 1$ indicating an assignment to the treatment group. Consider the $n^{th}$ individual in an experiment with a total of $N$ units (i.e. $n = 1, ..., N$). Let $\boldsymbol{z}_n(t_n) \in \mathcal{Z} \subset \mathbb{R}^d$ denote the potential value of the unobserved true mediator under the treatment assignment $t_n$. Since $Y$ depends on both $T$ and $Z$, we denote $y(t_n, \boldsymbol{z}_n(t_n)) \in \mathcal{Y} \subset \mathbb{R}$ as the potential outcome of the $n^{th}$ individual under treatment $t_n$ and true mediator $\boldsymbol{z}_n(t_n)$. Following Pearl (2001); Hicks & Tingley (2011); Robins & Greenland (1992), we define the average causal mediation effects (ACME), the average direct effects (ADE), and the average total effect (ATE) as

$$ACME(t) := \mathbb{E}\left[y(t, \boldsymbol{z}(1)) - y(t, \boldsymbol{z}(0))\right], \tag{1}$$

$$ADE(t) := \mathbb{E}\left[y(1, \boldsymbol{z}(t)) - y(0, \boldsymbol{z}(t))\right], \tag{2}$$

$$ATE := \mathbb{E}\left[y(1, \boldsymbol{z}(1)) - y(0, \boldsymbol{z}(0))\right], \tag{3}$$

where the expectations are taken over all the samples in our experiment. Our main objective is to recover these quantities as accurately as possible. As $\boldsymbol{z}_n$ is unobserved, we must infer $\boldsymbol{z}_n$ from the related observed feature $\boldsymbol{x}_n \in \mathcal{X} \subset \mathbb{R}^D$ with a much higher dimension, i.e. $D \gg d$, as well as any other available information. In practice, there often exists a set of observed covariates $\boldsymbol{w}_n \in \mathcal{W} \subset \mathbb{R}^m$ that also acts as confounders for $T$, $Z$, and $Y$ as shown in Figure 1b. With the presence of observed feature $\boldsymbol{x}_n$ and covariates $\boldsymbol{w}_n$, we make the following assumptions to make valid inferences about the causal effects:

**Assumption 1.** There exists an observed variable $X \in \mathcal{X} \subset \mathbb{R}^D$ that is caused by the unobserved true mediator $Z \in \mathcal{Z} \subset \mathbb{R}^d$ as shown in Figure 1.

**Assumption 2.** The following two conditional independence assumptions hold sequentially.

$$\{Y(t', z), Z(t)\} \perp\!\!\!\perp T | W = w, \tag{4}$$

$$Y(t', z) \perp\!\!\!\perp Z(t) | T = t, W = w, \tag{5}$$

where $0 < p(T = t | W = w) < 1$, $0 < p(Z(t) = z | T = t, W = w) < 1$, and $t, t' \in \{0, 1\}$.

Assumption 2 is first introduced by Imai et al. (2010), which is also known as *sequential ignorability*. Note that Equation 4 is equivalent to the strong ignorability assumption common in causal inference (Rosenbaum & Rubin, 1983; Rubin, 2005). It states that the treatment assignment $T$ is statistically independent of potential outcome $Y$ and potential mediators $Z$ given covariates $W$. Equation 5 states that given the treatment and covariates, the mediator $Z$ can be viewed as if it was randomized (in other words, there are no explained "backdoor" paths between the mediator and outcome (Pearl, 2014)).

### 3.1 Recovering Effects with Unobserved Mediator

One may question whether we can recover the direct, indirect, and total effects while $Z$ remains unobserved. When $X$ is used as a *proxy variable* for $Z$, we argue that this concern can be addressed by following the idea of "effect restoration" raised by Pearl (2012) in the context of discrete variables when the conditional distribution $p(\boldsymbol{x}|\boldsymbol{z})$ is correctly postulated and the following derivations can be naturally extended to the case of continuous variables by replacing the summations with integrals. To be specific, using the do-calculus developed by Pearl (1995), the indirect effect $p(y|do(\boldsymbol{z}), t)$, direct effect $p(y|do(t), \boldsymbol{z})$, and total effect $p(y|do(t))$ can be calculated using Bayesian network factorization:

$$p(y|do(t)) = \sum_{\boldsymbol{z}} p(y|\boldsymbol{z}, t)p(\boldsymbol{z}|t) = \sum_{\boldsymbol{z}} \frac{p(y, \boldsymbol{z}, t)}{p(t)}. \tag{6}$$

Since $X$ only depends on $Z$, i.e., $p(\boldsymbol{x}|y, \boldsymbol{z}, t) = p(\boldsymbol{x}|\boldsymbol{z})$, we can write:

$$p(y, t, \boldsymbol{x}) = \sum_{\boldsymbol{z}} p(y, \boldsymbol{z}, t, \boldsymbol{x}) = \sum_{\boldsymbol{z}} p(\boldsymbol{x}|y, \boldsymbol{z}, t)p(y, \boldsymbol{z}, t) = \sum_{\boldsymbol{z}} p(\boldsymbol{x}|\boldsymbol{z})p(y, \boldsymbol{z}, t). \tag{7}$$

For each pair of $y$ and $t$, we can interpret $p(\boldsymbol{x}|\boldsymbol{z})p(y, \boldsymbol{z}, t)$ as a matrix-vector multiplication:

$$V(\boldsymbol{x}) = \sum_{\boldsymbol{z}} M(\boldsymbol{x}, \boldsymbol{z})V(\boldsymbol{z}), \tag{8}$$

where $V(\boldsymbol{x}) = p(y, t, \boldsymbol{x})$, $V(\boldsymbol{z}) = p(y, \boldsymbol{z}, t)$, and $M(\boldsymbol{x}, \boldsymbol{z}) = p(\boldsymbol{x}|\boldsymbol{z})$ is a stochastic matrix (i.e., a square matrix whose columns are probability vectors). Under fairly broad conditions, $M(\boldsymbol{x}, \boldsymbol{z})$ has an inverse (call it $I(\boldsymbol{z}, \boldsymbol{x})$), which allows us to write:

$$p(y, \boldsymbol{z}, t) = \sum_{\boldsymbol{x}} I(\boldsymbol{z}, \boldsymbol{x})p(y, t, \boldsymbol{x}). \tag{9}$$

Therefore, we have:

$$p(y|do(t)) = \frac{1}{p(t)} \sum_{\boldsymbol{z}} \sum_{\boldsymbol{x}} I(\boldsymbol{z}, \boldsymbol{x})p(y, t, \boldsymbol{x}). \tag{10}$$

Note that $I(\boldsymbol{z}, \boldsymbol{x})$ is calculated on the conditional distribution $p(\boldsymbol{x}|\boldsymbol{z})$, which we will be learning in practice. In this way, we have expressed the total effect $p(y|do(t))$ in terms of the observed variables and known probability distributions.

Using a similar approach, we can also calculate the indirect effect $p(y|do(\boldsymbol{z}), t)$ and the direct effect $p(y|do(t), \boldsymbol{z})$. For the indirect effect, according to the second rule of do-calculus (Pearl, 1995), since $Y$ and $Z$ are $d$-separated in $G_Z$, where $G_Z$ denotes the causal graph in Figure 1a with all outgoing edges from node $Z$ removed, we have:

$$p(y|do(\boldsymbol{z}), t) = p(y|\boldsymbol{z}, t) = \frac{p(y, \boldsymbol{z}, t)}{p(\boldsymbol{z}, t)}. \tag{11}$$

Since equation 9 can be alternatively presented as $p(\boldsymbol{z}, t) = \sum_{\boldsymbol{x}} I(\boldsymbol{z}, \boldsymbol{x})p(t, \boldsymbol{x})$, we have:

$$p(y|do(\boldsymbol{z}), t) = \frac{\sum_{\boldsymbol{x}} I(\boldsymbol{z}, \boldsymbol{x})p(y, t, \boldsymbol{x})}{\sum_{\boldsymbol{x}} I(\boldsymbol{z}, \boldsymbol{x})p(t, \boldsymbol{x})}. \tag{12}$$

With this, we have shown that both the indirect effect and the total effect are identifiable in terms of the distributions of observed variables. Under the assumption of no interactions between the treatment and the mediator, which is known as moderation (Pearl & Mackenzie, 2018), this guarantees that the direct effect is also identifiable as the total effect is just a linear combination of the direct and indirect effects.

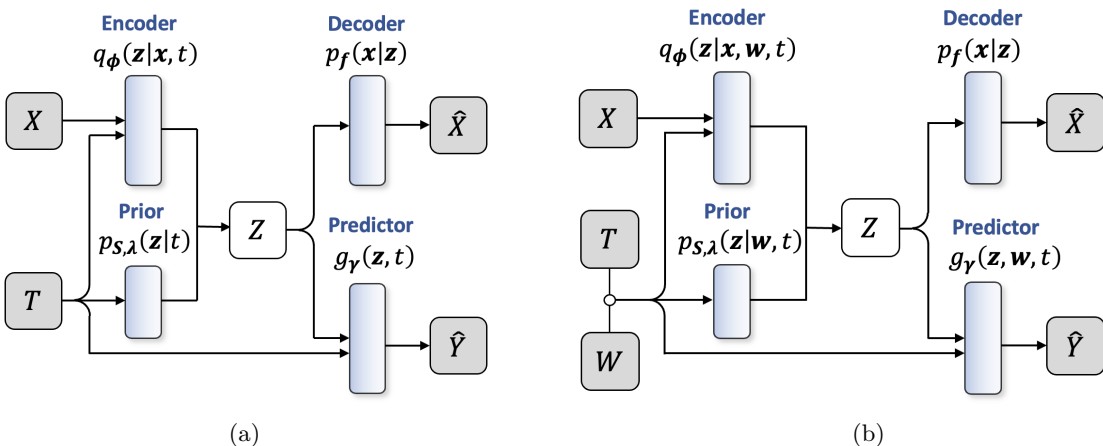

Figure 2: Illustration of the overall architecture of IMAVAE for (a) case without observed covariates and (b) case with observed covariates. Note that in case (b) the treatment assignment $T$ and the observed covariates $W$ are first concatenated and then passed into the prior, encoder, and decoder.

## 4 Method

We leverage the model structure of identifiable variational autoencoder (iVAE) (Khemakhem et al., 2020) to estimate the causal mediation effects based on the causal graphs illustrated in Figure 1. We believe the principled disentenglement of $p(z|t = 0)$ and $p(z|t = 1)$ introduced by iVAE can be helpful for accurate mediation effect estimation. However, we are not making any theoretical arguments here as reconstructing probabilities of the latent variables does not necessarily require disentanglement (Xi & Bloem-Reddy, 2023). In the following sections, we first present our framework in Section 4.1, and formally state the identifiability of our framework in Section 4.2.

### 4.1 Estimating Mediation Effect with VAE

The overall architecture of our framework, named Identifiable Mediation Analysis with Variational Autoencoder (IMAVAE), is illustrated in Figure 2, which consists of a variational posterior $q_\phi(z|x, u)$ and a conditional generative model $p_\theta(x, z|u) = p_{\mathbf{f}}(x|z)p_{S,\lambda}(z|u)$ where $\mathbf{f}$ is an injective function such that $p_{\mathbf{f}}(x|z) = p_\epsilon(x - \mathbf{f}(z))$ and $\epsilon$ is an independent noise variable with probability density function $p(\epsilon)$. Additionally, our framework incorporates a parametric model, denoted as $g_\gamma$, for the prediction of the outcome variable $\hat{Y}$. The prior distribution $p(z|u)$ is assumed to be conditionally factorial with each $z_i \in z$ belonging to a univariate exponential family as specified by the following probability density function:

$$p_{S,\lambda}(z|u) = \prod_i \frac{Q_i(z_i)}{C_i(u)} \exp \left[ \sum_{j=1}^{k} S_{i,j}(z_i)\lambda_{i,j}(u) \right], \tag{13}$$

where $Q_i$ is the base measure, $C_i(u)$ is the normalizing constant, $k$ is a pre-defined number of sufficient statistics, $S_i = (S_{i,1}, ..., S_{i,k})$ are the sufficient statistics, and $\lambda_i(u) = (\lambda_{i,1}(u), ..., \lambda_{i,k}(u))$ are the natural parameters. Figures 2a and 2b depict two variants of our framework, corresponding to the two cases outlined in the causal graphs in Figure 1:

- *Case (a)*: Without observed covariates, the treatment assignment $T$ is employed as the auxiliary variable and serves as input to the encoder, prior, and predictor, as illustrated in Figure 2a.

- *Case (b)*: With observed covariates, we first concatenate the observed covariates $W$ and the treatment assignment $T$. The concatenated vector $(W, T)$ is then passed into the encoder, prior, and predictor as the auxiliary variable, as illustrated in Figure 2b.

We denote the parameters of the encoder as $\phi$, parameters of the generative model as $\theta = \{\mathbf{f}, \mathbf{S}, \lambda\}$, and parameters of the predictor as $\gamma$. When fitting IMAVAE to the observed data, we optimize the parameter vector $(\theta, \phi, \gamma)$ by minimizing the following objective:

$$\theta^*, \phi^*, \gamma^* \coloneqq \arg\min_{\theta, \phi, \gamma} \left\{ \alpha \mathcal{L}_{\theta,\phi}^{\mathrm{RECON}}(\hat{\boldsymbol{x}}, \boldsymbol{x}) - \beta \mathcal{L}_{\theta,\phi}^{\mathrm{ELBO}}(\boldsymbol{x}, \boldsymbol{u}) + \mathcal{L}_{\phi,\mathbf{S},\lambda,\gamma}^{\mathrm{PRED}}(\hat{y}, y) \right\}, \tag{14}$$

where $\boldsymbol{u} = t$ for case (a), $\boldsymbol{u} = (\boldsymbol{w}, t)$ for case (b), $\mathcal{L}_{\theta,\phi}^{\mathrm{RECON}}(\hat{\boldsymbol{x}}, \boldsymbol{x})$ is the discrepancy between the input feature $\boldsymbol{x}$ and its reconstruction $\hat{\boldsymbol{x}}$, $\mathcal{L}_{\phi,\mathbf{S},\lambda,\gamma}^{\mathrm{PRED}}(\hat{y}, y)$ is the error between the predicted outcome $\hat{y}$ and the true outcome $y$, and $\mathcal{L}_{\theta,\phi}^{\mathrm{ELBO}}(\boldsymbol{x}, \boldsymbol{u})$ is the evidence lower bound (ELBO) with the following form:

$$\log p_\theta(\boldsymbol{x}|\boldsymbol{u}) \geq \mathcal{L}_{\theta,\phi}(\boldsymbol{x}, \boldsymbol{u}) \coloneqq \mathbb{E}_{q_\phi(\boldsymbol{z}|\boldsymbol{x},\boldsymbol{u})} \left[ \log p_\theta(\boldsymbol{x}, \boldsymbol{z}|\boldsymbol{u}) - \log q_\phi(\boldsymbol{z}|\boldsymbol{x}, \boldsymbol{u}) \right], \tag{15}$$

where we use the reparameterization trick to sample from $q_\phi(\boldsymbol{z}|\boldsymbol{x}, \boldsymbol{u})$. It is worth noting that there is some overlap in Equation 14 with the ELBO formulation in Equation 15, as both equations contain a reconstruction term. However, we adopt this specific formulation to emphasize the independence of each term while maintaining the overall loss through appropriately chosen weighting factors. The hyperparameters $\alpha$ and $\beta$ govern the relative importance assigned to the reconstruction error and the ELBO, respectively. In our experimental setup, we employ mean squared error (MSE) loss for both $\mathcal{L}_{\theta,\phi}(\hat{\boldsymbol{x}}, \boldsymbol{x})$ and $\mathcal{L}_{\phi,\mathbf{S},\lambda,\gamma}(\hat{y}, y)$. Furthermore, the prior distribution $p_{\mathbf{S},\lambda}(\boldsymbol{z}|\boldsymbol{u})$ is defined as a multivariate normal distribution, with its mean and covariance parameterized as functions of $\boldsymbol{u}$ through a neural network.

To give an estimation on the direct, mediated, and total effects after fitting the parameters, we repeatedly sample $\boldsymbol{z}(t)$ from the learned distributions (i.e., $p_{\mathbf{S},\lambda}(\boldsymbol{z}|t)$ for case (a) and $p_{\mathbf{S},\lambda}(\boldsymbol{z}|\boldsymbol{w}, t)$ for case (b)). Next, we feed both $\boldsymbol{z}(t)$ and the auxiliary variables into the predictor $g_\gamma$ to obtain $y(t, \boldsymbol{z}(t))$ for case (a) or $y(t, \boldsymbol{w}, \boldsymbol{z}(t))$ for case (b). Finally, we estimate the ACME, ADE, and ATE according to Equations 1-3 using estimated values of $y$.

## 4.2 Identifiability of IMAVAE

In this section, we prove the identifiability of IMAVAE by using similar definitions and assumptions stated by Khemakhem et al. (2020). Specifically, let $\mathcal{Z} \subset \mathbb{R}^d$ be the support of distribution of $\boldsymbol{z}$. The support of distribution of $\boldsymbol{u}$ is $\mathcal{U} = \{0, 1\}$ for case (a) and $\mathcal{U} = \{0, 1\} \times \mathcal{W} \subset \mathbb{R}^{m+1}$ for case (b). We denote by $\mathbf{S} \coloneqq (\mathbf{S}_1, ..., \mathbf{S}_d) = (S_{1,1}, ..., S_{d,k}) \in \mathbb{R}^{dk}$ the vector of sufficient statistics of Equation 13 and $\lambda(\boldsymbol{u}) = (\lambda_1(\boldsymbol{u}), ..., \lambda_d(\boldsymbol{u})) = (\lambda_{1,1}(\boldsymbol{u}), ..., \lambda_{d,k}(\boldsymbol{u})) \in \mathbb{R}^{dk}$ the vector of its parameters. Following the same notations in Khemakhem et al. (2020), we define $\mathcal{X} \subset \mathbb{R}^D$ as the image of $\mathbf{f}$ in Equation 13 and denote by $\mathbf{f}^{-1} : \mathcal{X} \to \mathcal{Z}$ the inverse of $\mathbf{f}$. Furthermore, we make the following assumption on the predictor:

**Assumption 3.** The predictor $g_\gamma(\boldsymbol{z}, \boldsymbol{u})$ takes the following form:

$$g_\gamma(\boldsymbol{z}, \boldsymbol{u}) \coloneqq p_{\mathbf{h}}(y|\boldsymbol{z}, \boldsymbol{u}) = p_\xi(y - \mathbf{h}(\boldsymbol{z}, \boldsymbol{u})), \tag{16}$$

where the function $\mathbf{h} : \mathcal{Z} \times \mathcal{U} \to \mathcal{Y}$ is injective, $\mathcal{Y} \subset \mathbb{R}$ is the image of $\mathbf{h}$, and $\xi$ is an independent noise variable with probability density function $p_\xi(\xi)$.

Similar to Khemakhem et al. (2020), for the sake of analysis, we treat $\mathbf{h}$ as a parameter of the entire model and define $\psi \coloneqq (\mathbf{f}, \mathbf{h}) : \mathcal{Z} \times \mathcal{U} \to \mathcal{X} \times \mathcal{Y}$. $\psi$ remains injective since both $\mathbf{f}$ and $\mathbf{h}$ are injective, and we consider the projection $\psi^{-1}$ on $\mathcal{Z}$ to be $\psi_{|z}^{-1}$. The domain of parameters is thus $\Theta = \{\theta \coloneqq (\mathbf{f}, \mathbf{h}, \mathbf{S}, \lambda)\}$. To formally present our claim, we give the following definitions:

**Definition 1.** Let $\sim$ be an equivalence relation on $\Theta$. We say that $p_\theta(\boldsymbol{x}, \boldsymbol{z}, y|\boldsymbol{u})$ is identifiable up to $\sim$ if $p_\theta(\boldsymbol{x}, \boldsymbol{z}, y|\boldsymbol{u}) = p_{\tilde{\theta}}(\boldsymbol{x}, \boldsymbol{z}, y|\boldsymbol{u}) \implies \theta \sim \tilde{\theta}$.

**Definition 2.** Let $\sim_A$ be the equivalence relation on $\Theta$ defined as follows:

$$\begin{aligned} (\mathbf{f}, \mathbf{h}, \mathbf{S}, \lambda) &\sim (\tilde{\mathbf{f}}, \tilde{\mathbf{h}}, \tilde{\mathbf{S}}, \tilde{\lambda}) \iff \\ &\exists A, \mathbf{c} \,|\, \mathbf{S}(\psi_{|z}^{-1}(\boldsymbol{x}, y)) = A\tilde{\mathbf{S}}(\tilde{\psi}_{|z}^{-1}(\boldsymbol{x}, y)) + \mathbf{c}, \\ &\forall \boldsymbol{x} \in \mathcal{X}; y \in \mathcal{Y}, \end{aligned} \tag{17}$$

where $A$ is an invertible $dk \times dk$ matrix and $\mathbf{c}$ is a vector.

With all the assumptions and definitions stated above, we state our theorem below as an extension of the results in Khemakhem et al. (2020). The detailed proof is in Appendix A.

**Theorem 1.** *(Extension to Theorem 1 in Khemakhem et al. (2020)) Assume that we observe data sampled from the generative model $p_{\boldsymbol{\theta}}(\boldsymbol{x}, \boldsymbol{z}, y|\boldsymbol{u}) = p_{\boldsymbol{f}}(\boldsymbol{x}|\boldsymbol{z})p_{\boldsymbol{h}}(y|\boldsymbol{z}, \boldsymbol{u})p_{\boldsymbol{S},\boldsymbol{\lambda}}(\boldsymbol{z}|\boldsymbol{u})$ where $p_{\boldsymbol{f}}(\boldsymbol{x}|\boldsymbol{z})$, $p_{\boldsymbol{h}}(y|\boldsymbol{z}, \boldsymbol{u})$ and $p_{\boldsymbol{S},\boldsymbol{\lambda}}(\boldsymbol{z}|\boldsymbol{u})$ follow the distributional form defined in Section 4.1, Equation 16, and Equation 13, respectively. Then the parameters $(\boldsymbol{f}, \boldsymbol{h}, \boldsymbol{S}, \boldsymbol{\lambda})$ will be $\sim_A$-identifiable if we assume the following holds:*

1. *The set $\{(\boldsymbol{x}, y) \in \mathcal{X} \times \mathcal{Y} \,|\, \varphi_{\boldsymbol{\epsilon}}(\boldsymbol{x}) = 0, \varphi_{\xi}(y) = 0\}$ has measure zero, where $\varphi_{\boldsymbol{\epsilon}}$ and $\varphi_{\boldsymbol{\xi}}$ are the characteristic functions of $p_{\boldsymbol{\epsilon}}$ and $p_{\boldsymbol{\xi}}$ defined in Section 4.1 and Equation 16, respectively.*

2. *The functions $\boldsymbol{f}$ and $\boldsymbol{h}$ are both injective.*

3. *The sufficient statistics $S_{i,j}$ in Equation 13 are differentiable almost everywhere, and $(S_{i,j})_{1 \leq j \leq k}$ are linearly independent on any subset of $\mathcal{Y}$ of measure greater than zero.*

4. *There exists $dk + 1$ distinct points $\boldsymbol{u}_0, ..., \boldsymbol{u}_{dk}$ such that the matrix $L = (\boldsymbol{\lambda}(\boldsymbol{u}_1) - \boldsymbol{\lambda}(\boldsymbol{u}_0), ..., \boldsymbol{\lambda}(\boldsymbol{u}_{dk}) - \boldsymbol{\lambda}(\boldsymbol{u}_0))$ of size $dk \times dk$ is invertible.*

We believe the first 3 assumptions in the aforementioned theorem are relatively easy to understand. The fourth assumption serves to introduce an additional degree of freedom into the relationship between the auxiliary variable $\boldsymbol{u}$ and the natural parameters $\boldsymbol{\lambda}$ of the prior distribution. This augmentation enables the utilization of $\boldsymbol{u}_0$ as a pivot, facilitating the demonstration of identifiability up to a linear transformation. In addition, we note that Theorem 1 generally holds for case (b). However, for case (a) with binary treatments, Theorem 1 only holds for a one-dimensional mediator, i.e., $\boldsymbol{z} \in \mathbb{R}$, as $\boldsymbol{u}$ only has two distinct values so $dk + 1 = 2$ has just one solution, which is $d = k = 1$. While we acknowledge that the fourth assumption is a bit strict, it is noteworthy that this set of assumptions finds application in other works, such as those by Hyvarinen et al. (2019) and Zhou & Wei (2020). Moreover, it is important to highlight that our extension of the identifiability theorem, originally presented in Khemakhem et al. (2020), incorporates the additional conditioning of $y$ on $\boldsymbol{u}$, thereby broadening the scope of iVAE.

## 5 Experiments

In accordance with prevailing practices in recent causal inference literature, we evaluate the performance of IMAVAE on a total of 4 datasets comprising 1 synthetic dataset and 3 semi-synthetic datasets[1], which facilitates benchmarking and comparative analysis of our results. It is worth noting that although we specify the predictor $g_{\boldsymbol{\gamma}}$ to be a conditional distribution reparameterized by function $\mathbf{h}$ in Section 4.2, in practice, we can simply use discriminative models such as linear/logistic regression (LR) or multi-layer perceptron (MLP) to estimate $g_{\boldsymbol{\gamma}}$, as demonstrated in the experiments conducted in Sections 5.1 and 5.2. When training IMAVAE to minimize the objective in Equation 15, we use the Adam optimizer and adopt parameter annealing so that the KL divergence will gradually come up to full strength in ELBO. All experiments are conducted on a computer cluster equipped with a GeForce RTX 2080 Ti GPU. The detailed experimental setup (e.g. training details, computing resources, licenses, etc.) is given in Appendix B.

### 5.1 Synthetic Dataset

To conduct an ablation study on hyperparameters $\alpha$ and $\beta$, we first construct a synthetic dataset following the causal graphs in Figure 1, where we model the outcome $Y$ to be linearly dependent on $T$, $Z$, and $W$. The details of data generation process are given in Appendix C. We set the unobserved true mediator to be two-dimensional (i.e. $d = 2$) for easier visualization. We display the distributions of the true and estimated unobserved mediator in Figure 3. It can be observed that IMAVAE effectively learns *disentangled representations* of $Z$ for the control and treatment groups in the latent space, up to trivial indeterminacies such as rotations and sign flips, for cases both with and without observed covariates. If we remove the reconstruction term (i.e. $\alpha = -1$ due to the overlap of reconstruction terms in Equation 15), the shape and

---

[1] Anonymous code is available at: `https://anonymous.4open.science/r/IMAVAE-557B/`.

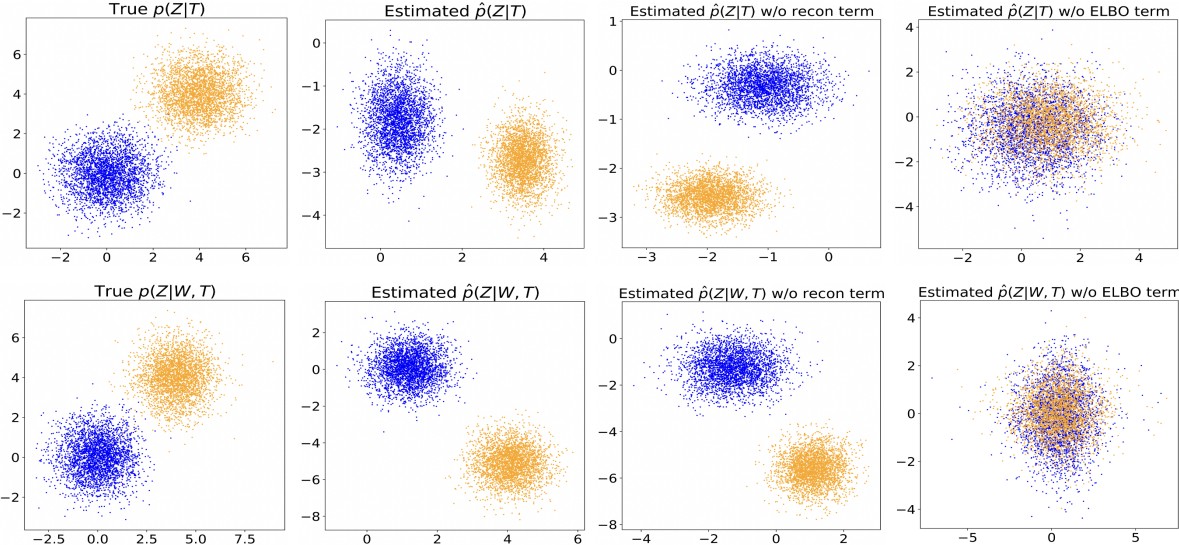

Figure 3: Distribution of the true and the estimated $p(\boldsymbol{z}|\boldsymbol{u})$ in the latent space where the upper row corresponds to case (a) without observed covariates, i.e. $\boldsymbol{u} = t$ and the bottom row corresponds for case (b) with observed covariates, i.e. $\boldsymbol{u} = (\boldsymbol{w}, t)$. From left to right, we present (left) the true distribution of $p(\boldsymbol{z}|\boldsymbol{u})$, (middle left) the estimated distribution $\hat{p}_{\boldsymbol{S},\boldsymbol{\lambda}}(\boldsymbol{z}|\boldsymbol{u})$ by IMAVAE, (middle right) the estimated distribution $\hat{p}_{\boldsymbol{S},\boldsymbol{\lambda}}(\boldsymbol{z}|\boldsymbol{u})$ without the reconstruction term, i.e. $\alpha = -1$, and (right) the estimated distribution $\hat{p}_{\boldsymbol{S},\boldsymbol{\lambda}}(\boldsymbol{z}|\boldsymbol{u})$ without the ELBO term, i.e. $\beta = 0$. The blue dots denote samples in control group and the orange dots denote samples in treatment group.

|  | IMAVAE (LR) | IMAVAE (MLP) | IMAVAE $\alpha = -1$ | IMAVAE $\beta = 0$ |
|---|---|---|---|---|
| ACME $(t = 1)$ | 0.056 $\pm$ .007 | 0.373 $\pm$ .007 | 0.078 $\pm$ .007 | 2.875 $\pm$ .016 |
| ADE $(t = 0)$ | 0.058 $\pm$ .000 | 0.370 $\pm$ .002 | 0.052 $\pm$ .000 | 0.043 $\pm$ .000 |
| ATE | 0.003 $\pm$ .007 | 0.003 $\pm$ .007 | 0.025 $\pm$ .006 | 2.917 $\pm$ .015 |

Table 1: Absolute error of ACME under treated, ADE under control, and ATE on the synthetic dataset for IMAVAE in case (a) *without* observed covariates

|  | IMAVAE (LR) | IMAVAE (MLP) | IMAVAE $\alpha = -1$ | IMAVAE $\beta = 0$ |
|---|---|---|---|---|
| ACME $(t = 1)$ | 0.214 $\pm$ .016 | 0.058 $\pm$ .016 | 0.379 $\pm$ .014 | 5.912 $\pm$ .028 |
| ADE $(t = 0)$ | 0.194 $\pm$ .000 | 0.032 $\pm$ .002 | 0.385 $\pm$ .000 | 0.127 $\pm$ .000 |
| ATE | 0.019 $\pm$ .015 | 0.027 $\pm$ .014 | 0.011 $\pm$ .016 | 6.036 $\pm$ .028 |

Table 2: Absolute error of ACME under treated, ADE under control, and ATE on the synthetic dataset for IMAVAE in case (b) *with* observed covariates

orientation of the distributions become slightly different but remain disentangled. However, if we discard the ELBO term (i.e. $\beta = 0$), the model fails to separate the distributions of control and treatment groups. We also compute the absolute errors between the estimated ACME, ADE, ATE, and their corresponding ground truths as shown in Tables 1 and 2. It can be observed that IMAVAE yields slightly larger errors when the reconstruction term is removed (i.e., $\alpha = -1$). When employing an MLP for $g_{\boldsymbol{\gamma}}$, we observe a slight increase in error for case (a) but a notable decrease in error for case (b). This discrepancy might be attributed to the fact that $\boldsymbol{w}$ has a much higher dimensionality than $t$, which requires higher model complexity. However, without the ELBO term (i.e., $\beta = 0$), the model produces significantly larger errors on ACME and ATE. From the obtained results, we conclude that the ELBO term is essential for learning a better representation of the unobserved mediator, which, in turn, improves the accuracy of mediation effect estimation.

### 5.1.1 Comparison to Conventional VAE

To demonstrate the necessity of conditioning $\boldsymbol{z}$ on $\boldsymbol{u}$, we do another ablation study by removing the conditioning of $\boldsymbol{z}$ on $\boldsymbol{u}$ for both the prior and the encoder, thereby transforming the backbone network into a

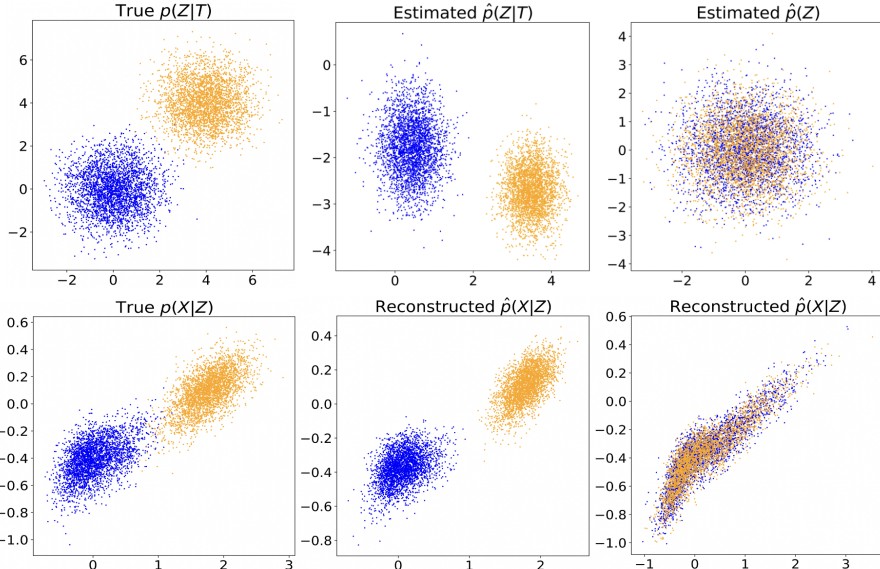

Figure 4: Comparison of IMAVAE with conventional VAE for case (a) without observed covariates. From left to right, we present (left) the true distribution, (middle) the estimated distribution by IMAVAE, and (right) the estimated distribution by conventional VAE. The top row corresponds to the prior distribution of $z$ in the latent space. The bottom row corresponds to the distribution of the first 2 dimensions of the reconstructed feature $x$.

conventional VAE. Subsequently, we repeat the experiment for both case (a) and case (b), as outlined in Section 5.1. In such a case, the link between the latent space and $u$ is broken. The results are presented in Table 3, which shows a notable degradation in model performance when the additional conditioning on $u$ is removed (i.e., conventional VAE) for both cases. Additionally, we visualize the prior distribution of $z$ in the latent space and the first two dimensions of the distribution of the reconstructed $x$ for case (a) in Figure 4. The visualization indicates that the reconstruction of $x$ by IMAVAE is much closer to the ground truth compared to that of the conventional VAE. Therefore, we argue that the conditioning of $u$ on $z$ is crucial for both accurate estimation of the mediation effect and better reconstruction of the observed feature.

## 5.2 Electrophysiological Dataset

**Data** As described in the introduction, causal mediation analysis holds significant relevance for applications in systems neuroscience. One such area is in the emerging area of targeted neurostimulation (see Deisseroth (2011); Limousin & Foltynie (2019); Tufail et al. (2011) for a description), where the brain is manipulated by optical, electrical, or mechanical stimulation with the goal of manipulating behavior in many brain conditions. Accurate appraisal of neural changes causing behavioral change will provide a deeper understanding of mechanisms driving neural

|  | Case (a) | | Case (b) | |
|---|---|---|---|---|
|  | IMAVAE | Conventional VAE | IMAVAE | Conventional VAE |
| ACME $(t=1)$ | 0.056 ± 0.007 | 3.153 ± 0.024 | 0.214 ± 0.016 | 6.321 ± 0.042 |
| ADE $(t=0)$ | 0.058 ± 0.000 | 0.346 ± 0.000 | 0.194 ± 0.000 | 1.243 ± 0.000 |
| ATE | 0.003 ± 0.007 | 2.809 ± 0.024 | 0.019 ± 0.015 | 5.073 ± 0.033 |

Table 3: Absolute error of ACME under treated, ADE under control, and ATE on the MNIST dataset for IMAVAE and other benchmarks.

activity and potentially lead to more efficacious treatments. We demonstrate capability of our method to this domain with a semisynthetic dataset by post-processing real multi-site brain recordings using local field potential (LFP) data from 26 mice (Gallagher et al., 2017), which is publicly available (Carlson et al., 2023). Specifically, we take the LFP signals as the observed feature $X$ and generate $Z$ by applying principal component analysis (PCA) on $X$. The outcome $Y$ is manually constructed as a linear function of the treatment $T$, the mediator $Z$, and the genotype $W$ (only for case (b) with observed covariate). The detailed procedure of dataset generation is given in Appendix D.

Table 4: Absolute error of ACME under treated, ADE under control, and ATE on the tail suspension test dataset for IMAVAE and other benchmarks.

| | Case (a) | | | | | Case (b) | | | | |
|---|---|---|---|---|---|---|---|---|---|---|
| | Shallow LSEM | Deep LSEM | HIMA | IMAVAE (LR) | IMAVAE (MLP) | Shallow LSEM | Deep LSEM | HIMA | IMAVAE (LR) | IMAVAE (MLP) |
| ACME $(t=1)$ | 15.14 $\pm 0.07$ | 15.48 $\pm 0.07$ | 13.95 $\pm 0.03$ | **2.348** $\pm$ **0.003** | 3.849 $\pm 0.002$ | 3.06 $\pm 2.16$ | 4.80 $\pm 0.44$ | 2.67 $\pm 0.02$ | 0.559 $\pm 0.002$ | **0.164** $\pm$ **0.002** |
| ADE $(t=0)$ | 14.71 $\pm 0.03$ | 15.06 $\pm 0.07$ | 3.82 $\pm 0.01$ | **1.603** $\pm$ **0.000** | 3.406 $\pm 0.004$ | 5.03 $\pm 1.20$ | 5.16 $\pm 0.51$ | 3.21 $\pm 0.01$ | 0.782 $\pm 0.000$ | **0.067** $\pm$ **0.000** |
| ATE | **0.42** $\pm$ **0.06** | 0.42 $\pm 0.14$ | 17.77 $\pm 0.03$ | 0.744 $\pm 0.003$ | 0.443 $\pm 0.002$ | 1.96 $\pm 3.36$ | 0.36 $\pm 0.94$ | 0.54 $\pm 0.02$ | 0.223 $\pm 0.002$ | **0.097** $\pm$ **0.002** |

**Results**  We compare our method with two baseline models that are designed to handle high-dimensional mediators: an integrated framework of shallow or deep neural network and linear SEM (Shallow/Deep LSEM) (Nath et al., 2023) and a high-dimensional mediation analysis (HIMA) framework (Zhang et al., 2016). In the shallow/deep LSEM frameworks, the high-dimensional mediator is initially mapped onto a low-dimensional space, and subsequently, the coefficients of a linear SEM are fitted using this low-dimensional mediator. Additionally, HIMA is implemented as an R package which considers each component of $Z$ as an individual mediator instead of a multidimensional mediator. As such, we report the mediation effect using the component with the highest correlation. We compute and display the absolute errors of ACME, ADE, and ATE in Table 4. Our results indicate that IMAVAE outperforms both benchmarks by a very wide margin on all estimations except the ATE in case (a) without covariates. When using MLP for $g_{\gamma}$, we observe a similar phenomenon as discussed in Section 5.1 where IMAVAE yields slightly larger error in case (a) while exhibits improved performance in case (b). The two benchmarks used in this experiment yield significantly larger errors on ACME and ADE. We believe this is reasonable, as both benchmarks are designed based on linear SEMs and are thus not able to capture the correlation between the components of $Z$.

## 5.3  Concatenated MNIST Dataset

To further illustrate IMAVAE's ability to handle high-dimensional mediators, we conducted an additional experiment involving simulations using digit images from the MNIST dataset. This data generation procedure closely follows the methodology outlined in simulation 1 of Nath et al. (2023). The major modification is that we sample $T$ as a binary stimulus, deviating from the original

| | IMAVAE (ours) | Deep LSEM | Shallow LSEM | SVR |
|---|---|---|---|---|
| ACME $(t=1)$ | $0.273 \pm 0.4820$ | $0.281 \pm 0.2710$ | $0.378 \pm 0.4050$ | $1.286 \pm 0.0001$ |
| ADE $(t=0)$ | $0.051 \pm 0.0000$ | $0.139 \pm 0.2730$ | $0.042 \pm 0.4110$ | $0.867 \pm 0.0001$ |
| ATE | $0.421 \pm 0.3699$ | $0.419 \pm 0.5441$ | $0.420 \pm 0.8160$ | $0.419 \pm 0.0003$ |

Table 5: Absolute error of ACME under treated, ADE under control, and ATE on the MNIST dataset for IMAVAE and other benchmarks.

process where $T$ was sampled from a standard normal distribution. Our simulation emulates a scenario in which the observed data $X$ is a complex and nonlinear function of a set of mediator variables. To achieve this, we first take the manually constructed mediator $Z$ and and compute its cumulative distribution function of a normal distribution (i.e., $norm.cdf(Z)$). Then we extract the first 4 digits after the decimal point of $norm.cdf(Z)$, randomly choose corresponding images of these digits in the MNIST dataset, and generate $X$ by concatenating them into a larger image. The detailed simulation procedure is elaborated in Appendix E.

We compute and display the absolute errors of ACME, ADE, and ATE in Table 5 and other benchmarks, including SEMs with a deep neural network and a shallow neural network (Nath et al., 2023), and a support vector regression (SVR). It can be observed that IMAVAE achieves the lowest error on estimating the mediation effect, while demonstrating comparable performance in estimating the direct and total effects.

Table 6: Absolute error of ACME under treated, ADE under control, and ATE on simulated Jobs II data for IMAVAE and other benchmarks where 10% of the data are mediated (i.e. $Z > 3$).

| | LSEM-I | | NEM-I | | IPW | | CMAVAE | | IMAVAE (LR) | |
|---|---|---|---|---|---|---|---|---|---|---|
| $N$ | 500 | 1000 | 500 | 1000 | 500 | 1000 | 500 | 1000 | 500 | 1000 |
| | ACME under treated ($t = 1$) | | | | | | | | | |
| $\eta = 10$ | .90 ± .04 | .60 ± .02 | .60 ± .03 | .80 ± .01 | .60 ± .04 | .80 ± .02 | .20 ± .00 | .30 ± .00 | **.12 ± .02** | **.10 ± .01** |
| $\eta = 1$ | .00 ± .01 | .10 ± .01 | **.00 ± .00** | .10 ± .01 | .00 ± .01 | .10 ± .01 | .10 ± .00 | .10 ± .00 | .07 ± .01 | **.05 ± .01** |
| | ADE under control ($t = 0$) | | | | | | | | | |
| $\eta = 10$ | 1.3 ± .07 | 1.6 ± .06 | 1.2 ± .06 | 1.8 ± .05 | 1.2 ± .06 | .20 ± .06 | **.10 ± .00** | **.00 ± .03** | .33 ± .00 | .32 ± .00 |
| $\eta = 1$ | 3.3 ± .08 | **.00 ± .07** | 1.1 ± .03 | .20 ± .07 | 3.3 ± .08 | .30 ± .06 | .50 ± .02 | .40 ± .01 | **.25 ± .00** | .27 ± .00 |
| | ATE | | | | | | | | | |
| $\eta = 10$ | 2.2 ± .05 | 1.0 ± .06 | 1.8 ± .05 | .90 ± .06 | .50 ± .05 | 1.0 ± .06 | .30 ± .01 | .30 ± .03 | **.21 ± .02** | **.23 ± .01** |
| $\eta = 1$ | 3.3 ± .08 | .10 ± .07 | 3.4 ± .03 | **.10 ± .06** | 3.2 ± .07 | .20 ± .05 | .40 ± .02 | .30 ± .01 | **.18 ± .01** | .22 ± .01 |

Table 7: Absolute error of ACME under treated, ADE under control, and ATE on simulated Jobs II data for IMAVAE and other benchmarks where 50% of the data are mediated (i.e. $Z > 3$)

| | LSEM-I | | NEM-I | | IPW | | CMAVAE | | IMAVAE (LR) | |
|---|---|---|---|---|---|---|---|---|---|---|
| $N$ | 500 | 1000 | 500 | 1000 | 500 | 1000 | 500 | 1000 | 500 | 1000 |
| | ACME under treated ($t = 1$) | | | | | | | | | |
| $\eta = 10$ | .90 ± .03 | .60 ± .03 | .20 ± .03 | .40 ± .03 | .20 ± .03 | .40 ± .03 | **.00 ± .00** | .10 ± .00 | .01 ± .05 | **.00 ± .03** |
| $\eta = 1$ | .10 ± .01 | **.00 ± .01** | .20 ± .00 | .10 ± .01 | .10 ± .01 | **.00 ± .01** | .10 ± .00 | .10 ± .00 | **.01 ± .05** | .01 ± .03 |
| | ADE under control ($t = 0$) | | | | | | | | | |
| $\eta = 10$ | .60 ± .06 | .10 ± .04 | .10 ± .06 | .10 ± .04 | .70 ± .07 | .20 ± .05 | .30 ± .01 | .10 ± .00 | **.09 ± .00** | **.08 ± .00** |
| $\eta = 1$ | .10 ± .10 | .30 ± .10 | .10 ± .10 | .30 ± .04 | .30 ± .10 | .20 ± .04 | .10 ± .00 | .10 ± .00 | **.09 ± .00** | **.09 ± .00** |
| | ATE | | | | | | | | | |
| $\eta = 10$ | .30 ± .05 | .80 ± .03 | .10 ± .05 | .50 ± .03 | .90 ± .05 | .20 ± .04 | .30 ± .01 | **.00 ± .01** | **.08 ± .04** | .07 ± .03 |
| $\eta = 1$ | .10 ± .09 | .30 ± .04 | .30 ± .10 | .30 ± .04 | .20 ± .10 | .20 ± .04 | **.00 ± .01** | .20 ± .01 | .08 ± .04 | **.08 ± .03** |

### 5.4 Jobs II Dataset

**Data**  To evaluate whether our method can generalize to real-world scenarios used in recent causal mediation analysis frameworks, we test IMAVAE on the Jobs II dataset Vinokur et al. (1995). To obtain the ground truth for direct and mediated effects, we followed a simulation procedure similar to Huber et al. (2016) to make ACMEs, ADEs, and ATE all equal to zero. The detailed simulation procedure is given in Appendix F.

**Results**  We compare the performance of our method with several benchmarks: nonlinear SEM with interaction (LSEM-I) (Imai et al., 2010), imputing-based natural effect model (NEM-I) (Lange et al., 2012), IPW (Huber et al., 2013), and Causal Mediation Analysis with Variational Autoencoder (CMAVAE) (Cheng et al., 2022). It is worth noting that the Jobs II dataset presents an observable mediator variable $Z$, which is *not* the optimal scenario for our proposed framework, as IMAVAE is specifically designed for CMA with *implicitly* observed mediators. Nonetheless, according to the results shown in Tables 6 and 7 (where $N$ is the total number of simulated samples and $\eta$ is a simulation parameter), our method still mostly outperforms the benchmarks in terms of the estimation on ACME, ADE, and ATE with a reasonable level of uncertainty.

## 6 Discussion

**Design Choice of the Predictor**  In Section 5, we have experimented with two choices of architecture for $g_{\gamma}$. From the empirical results, we notice that the MLP architecture typically works better in cases where we have covariates with higher dimensions. In practice, we can explore more flexible architectures for $g_{\gamma}$ to account for more complex causal relationships. Additionally, when using a simple linear/logistic regression for $g_{\gamma}$, IMAVAE gives almost deterministic predictions on the direct effects (as can be observed in the standard deviation of ADE errors). This can be explained by our inference procedure of direct and indirect effects. As elaborated at the end of Section 4.1, we estimate ACME and ADE to be $\mathbb{E}_{\boldsymbol{z} \sim p(\boldsymbol{z}|\boldsymbol{u})}[g_{\gamma}(t, \boldsymbol{z}(1)) - g_{\gamma}(t, \boldsymbol{z}(0))]$ and $\mathbb{E}_{\boldsymbol{z} \sim p(\boldsymbol{z}|\boldsymbol{u})}[g_{\gamma}(1, \boldsymbol{z}(t)) - g_{\gamma}(0, \boldsymbol{z}(t))]$, respectively. Note that in the calculation of ADE, we pass $\boldsymbol{z}(t)$ into $g_{\gamma}$ for both $T = 0$ and $T = 1$, thereby greatly reducing the uncertainty of ADE compared to that of ACME.

However, as we switch to a more complex model by using an MLP for $g_{\gamma}$, the uncertainty from the sampling of $z$ might be amplified.

**Limitations**   In Section 3, we have discussed the consideration of $X$ as a proxy variable in our causal graph. Nevertheless, as pointed out by Pearl (2012), incorrect postulation of $p(x|z)$ can lead to bias in the estimation of ACME, ADE, and ATE. This can occur, for example, when measuring LFP signals from an incorrect brain region in the mice, as discussed in Section 5.2. Even if $p(x|z)$ is accurately postulated, the presence of measurement error can increase sampling variability. Also, in Section 4.2, we prove the identifiability of some parts of IMAVAE under a collection of assumptions, including the absence of interactions between the treatment $T$ and the mediator $Z$. It is important to note, however, that this proof does not ensure the precise recovery of the true distribution $p(z|t)$. Nevertheless, the learned distribution can still facilitate accurate estimations of both direct and indirect effects. Furthermore, in cases when the dimension of $W$ is very high, it is advisable to first map $W$ into a lower dimension so that the representation power of $T$ will not be too low after concatenation.

**Applications and Broader Impacts**   We believe the proposed model architecture can be very useful for improving interpretability for neuroscience applications. For instance, the disentangled mediator representations obtained by IMAVAE can be used to investigate the brain activities of individuals under different interventions. It can also be combined with other interpretable methods such as linear factor models to better illustrate the high-dimensional dynamics in brain networks as proposed by Talbot et al. (2020).

## 7   CONCLUSION

This work makes a contribution to the field of causal mediation analysis (CMA) by proposing a novel method, IMAVAE, that can handle situations where the mediator is indirectly observed and observed covariates are likely to be present. Our approach builds on existing CMA methods and leverages the identifiable variational autoencoder (iVAE) model architecture to provide a powerful tool for estimating direct and mediated effects. We have demonstrated the effectiveness of IMAVAE in mediation effect estimation through theoretical analysis and empirical evaluations. Specifically, we have proved the identifiability of the joint distribution learned by IMAVAE and demonstrated the disentanglement of mediators in control and treatment groups. Overall, our proposed method offers a promising avenue for CMA in settings with much more complex data, where traditional methods may struggle to provide accurate estimates.

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
