# OpenReview forum: "Causal Mediation Analysis with Multi-dimensional and Indirectly Observed Mediators"
_TMLR — Rejected by TMLR_

### Review · Reviewer_egSw · 2023-12-18

**Summary Of Contributions:**

The authors present an approach to causal mediation analysis that uses deep generative learning, more concretely variational autoencoders, in order to model causal treatment effects in causal graphs with multi-dimensional and indirectly observed mediators. Based on assumptions building on prior work the authors show proofs demonstrating that the joint distribution of the observed and latent variables is identifiable in the proposed framework. Experiments on synthetic and semi-synthetic data sets are conducted.

**Audience:**

Yes

**Broader Impact Concerns:**

Estimating causal influences from observational data is of high relevance for policy makers and can have long ranging impacts on society. When establishing new methodology that could be used to determine causal influence of certain factors, authors should more clearly highlight the limitations and strengths of their method. As mentioned above under *weaknesses* the authors could improve the manuscript with respect to that aspect.

**Claims And Evidence:**

No

**Requested Changes:**

For a better assessment of the pros and cons of the proposed method, it would be great if the authors could:

* work on a more intutive and convincing motivation for the proposed approach. Even if it is novel to use this particular model with a multivariate latent variable approach, it would be easier to convince readers if the authors would also have a worked example and empirical evidence that demonstrates why the proposed approach makes sense - and why for instance supervised learning settings or other methods are less preferable

* clarify more clearly (ideally with synthetic data?) the limitations and unique selling points of the proposed methods, with respect to the main differences (multivariate latent variables subspaces?)

* clarify the above aspects on the empirical results

**Strengths And Weaknesses:**

# Strengths
* I think the authors make a valuable contribution, there is need for flexible non-linear causal modelling tools that go beyond linear SEMs
* The proofs look sound to the best of my knowledge
* It is great that the authors use both synthetic and surrogate data

# Weaknesses
* The theoretical side of the identifiability is sound, as far as I can see. What is not entirely clear to me when reading the manuscript is the motivation for this particular model and how the implicit assumptions of the particular model translate into the practical relevance of the proposed framework. In other words there were a couple questions that I wasn't sure I'd know the answers to when explaining this paper to others:
    * Why do we need a generative model in the first place, wouldn't standard supervised models work as well for this purpose?
    * What's the advantage of multivariate latent variables?
    * How are the latent variables interpretable? I mean, there could be many different equivalent LVMs that lead to the same results in terms of reconstruction error or other metrics, right?
    * Why not isotropic gaussians for the latent variables, would that be too restrictive for the proposed framework?

* While the experimental section has many positive aspects (many and diverse data sets, careful analyses), it seems that there are some things that could be clarified / improved:
    * the only case in which we can really assess the properties of the proposed method is the synthetic data set but in this case, no comparison is made with other methods.
    * in the case of the electrophysiology data there seems to be a very broad spectrum of results - but then there is a standard deviation of 0.000 for the proposed method in some cases? The mean metrics are some orders of magnitude off between the methods - this suggest a systematic disadvantage of some method in terms of preprocessing, hyperparameters, modelling assumptions (like: using the default parameters optimized for another application)
    * in the MNIST case: are these bold-faced results actually significantly different from the others? if not, it would be better to not bold-face any of those

---

> ### Author Response · Authors · 2024-01-10
> **Response to Reviewer egSw (1/2)**
>
> Thank you so much for your constructive comments. Regarding your concerns under weaknesses and questions, we have listed our answers below.
>
> **Why do we need a generative model in the first place, wouldn't standard supervised models work as well for this purpose?**
>
> The main objective of using a generative model is to learn a representation of the mediator in the latent space when the true mediator is indirectly or partially observed. While it is possible to estimate the mediation effect using a conventional supervised model, such an approach does not facilitate the learning of a latent representation for the mediator.
>
> **What's the advantage of multivariate latent variables?**
>
> As stated in the introduction, we believe multivariate latent variables can be useful for cases involving complex data such as neuroimaging and electrophysiology. For example, due to the high-dimensional and complex nature of the brain data, to precisely evaluate neural stimulations that cause behavioral change, we may need to use multivariate latent variables to represent such stimulations (e.g., optical, electrical, or mechanical stimulations) in the brain.
>
> **How are the latent variables interpretable? I mean, there could be many different equivalent LVMs that lead to the same results in terms of reconstruction error or other metrics, right?**
>
> We agree with the reviewer that there could be many other LVMs that lead to the same results. However, we would like to note that one of the key goals of iVAE and nonlinear ICA is to find independent components in data, particularly brain signals.  Likewise, we want to find independent components that mediate treatment responses, hence are applying also to neural signals. We note that much of the relevant work we cite on neural signals is focused on finding unique brain networks, much like the nonlinear ICA work.
>
> **Why not isotropic Gaussians for the latent variables, would that be too restrictive for the proposed framework?**
>
> Isotropic Gaussian is definitely an appropriate choice for the prior distribution of latent variables as it belongs to the exponential family. We follow the statement of Khemakhem et al. [1] which assumes that the $p(boldsymbol{z}|\boldsymbol{u})$ is conditionally factorial and each element of $z_i \in \boldsymbol{z}$ has a univariate exponential family distribution. This is not a very restrictive assumption due to the universal approximation capabilities of the exponential family distributions [2].
>
> [1]. Khemakhem, I., Kingma, D., Monti, R., & Hyvarinen, A. (2020, June). Variational autoencoders and nonlinear ica: A unifying framework. In International Conference on Artificial Intelligence and Statistics (pp. 2207-2217). PMLR.
>
> [2]. Sriperumbudur, B., Fukumizu, K., Gretton, A., Hyv, A., & Kumar, R. (2017). Density estimation in infinite dimensional exponential families. Journal of Machine Learning Research, 18(57), 1-59.
>
> **The only case in which we can really assess the properties of the proposed method is the synthetic data set, but in this case, no comparison is made with other methods.**
>
> The main objective of including a synthetic dataset is to perform an ablation study on different terms in the loss function (Equation 14) instead of assessing the relevant properties of the proposed method. We believe the most important property of our method is to learn a latent representation that facilitates accurate dissection of direct and indirect effects, and this property can be comprehensively evaluated on semi-synthetic datasets, as elaborated in Sections 5.2 to 5.4.

---

> ### Author Response · Authors · 2024-01-10
> **Response to Reviewer egSw (2/2)**
>
> **In the case of the electrophysiology data there seems to be a very broad spectrum of results - but then there is a standard deviation of 0.000 for the proposed method in some cases. The mean metrics are some orders of magnitude off between the methods - this suggests a systematic disadvantage of some methods in terms of preprocessing, hyperparameters, and modeling assumptions.**
>
> The standard deviation of 0.000 mainly appears in the ADE error, but this does not mean that the estimation of direct effect by our proposed method is deterministic. As requested by Reviewer 71Nt, we have included some additional empirical results by using a multi-layer perceptron for $g_{\gamma}$, and the standard deviation in that case is not 0.000 anymore. We pertain this phenomenon to our inference procedure of direct and indirect effects. Specifically, we estimate ACME and ADE to be $\mathbb{E}\_{\boldsymbol{z} \sim p(\boldsymbol{z}|\boldsymbol{u})} [g_{\boldsymbol{\gamma}}(t,\boldsymbol{z}(1)) - g_{\boldsymbol{\gamma}}(t, \boldsymbol{z}(0))]$ and $\mathbb{E}\_{\boldsymbol{z} \sim p(\boldsymbol{z}|\boldsymbol{u})} [g_{\boldsymbol{\gamma}}(1,\boldsymbol{z}(t)) - g_{\boldsymbol{\gamma}}(0, \boldsymbol{z}(t))]$, respectively. Note that in the calculation of ADE, we pass $\boldsymbol{z}(t)$ into $g_{\boldsymbol{\gamma}}$ for both $T = 0$ and $T = 1$, thereby greatly reducing the uncertainty of ADE compared to that of ACME. We have also included some relevant discussions in Section 6 of the revised manuscript.
>
> Regarding the experimental results obtained from benchmark methods, we systematically explored various data preprocessing procedures and hyperparameter settings, ensuring that the reported results represent the best outcomes achievable with these benchmark methods in our experiments. Our approach involved a comprehensive literature review to identify the most suitable CMA frameworks capable of effectively handling high-dimensional mediators. Consequently, the selected methods have been deemed the most appropriate choices for our study.
>
> **In the MNIST case: are these bold-faced results actually significantly different from the others? if not, it would be better to not bold face any of those.**
>
> Thank you for your suggestion. We have updated the MNIST table accordingly.

---

### Review · Reviewer_71Nt · 2023-12-21

**Summary Of Contributions:**

The manuscript proposes method for estimating mediated causal effects for cases where a possibly multivariate latent variable mediating the effect is observed indirectly. The solution builds on identifiable variational autoencoder that has theoretical identifiability guarantees, and in this work those guarantees are extended for the CMA scenario. The method is demonstrated on a collection of (semi)-synthetic data sets and shown to in many cases reduce the absolute error in estimating the effect.

**Audience:**

Yes

**Broader Impact Concerns:**

There are no direct broader impact concerns, but the paper needs to be clear in terms of communicating the limitations to avoid giving the impression the paper delivers a solution with provable guarantees of inferring true causal relationships. This is needed to avoid follow-up work that falsely trusts on the findings based on the theoretical analysis.

**Claims And Evidence:**

No

**Requested Changes:**

The paper needs substantial changes to satisfy the evaluation criteria.

1. The discussion on the effect of $g_{\gamma}$ being a simple model should be extended and the claims related to that need to be validated. You need to say how this choice relates to the theoretical analysis of identifiability of $z$ and you need to either remove the claim of simple $g_{\gamma}$ being enough or validate it empirically, by showing what happens if you use a more flexible network also in that part.

2. The claims like "*we prove that the true joint distribution over observed and latent variables is identifiable with the proposed method*" should be re-phrased to make it clear that you have a theoretical proof of certain characteristics under a collection of assumptions (that are quite strong and will not hold in the kind of examples you use for motivating the work) but cannot prove that *the proposed computaional method* would actually identify the correct latent variables.

3. The empirical experimentation needs to be made self-contained. Now many of the experimental details are only in Appendix and the reader cannot understand the experiments without consulting it, and many of the details needed for drawing conclusions are not provided at all (e.g. the annealing process or the network structures in Appendix B, or explanation of "*we construct the outcome $y$ manually*" in Appendix D). For instance, you never tell in the main paper that for the simulated data you use a linear model for $y$. It's not a surprise that your method works well with simple $g_{\gamma}$ if you only evaluate it in cases where the true mapping is linear, but the reader never learns this unless they read through equations in the Appendix in detail.


Some minor comments that can additionally be considered:
- Concatenation of $W$ and $T$ is technically correct, but likely to fail in practice when $W$ is very high-dimensional. There could be a comment on that and in general it reduces the value of covering the case with observed covariates separately since you only make the minimal effort of incorporating $W$ into your model, rather than developing a new scientifically interesting solution for that scenario.
- Figure 3 is blurry; please use higher resolution or a vector format
- How were $\alpha$ and $\beta$ chosen for the full model in the experiments?
- Appendix overloads some symbols; for example $\alpha$ is re-used in different meaning in simulation experiments.
- In Tables 3 and 4 your own method is the first column but in Tables 5-6 it is the last. Being consistent with the ordering would make the paper easier to read.

**Strengths And Weaknesses:**

The proposed approach appears in general sound and the paper is written in clear language and is easy to understand, and the topic is definitely interesting for the TMLR audience. The problem setup is not novel but the authors clearly explain how the previous methods make simplifying assumptions (linear relationships, low-dimensional latent variables) and a solution relaxing these is a useful contribution for the literature. The solution is relatively straightforward in the sense that existing VAE variant is used but the choice is well made. The variant is one that has identifiability guarantees and the authors are able to extend them for the CMA case, at least in some sense.

However, I have one major concern about the approach. In the empirical experimentation you make the simplifying assumption that $g_{\gamma}$, the mapping from $z$ to $x$ is a (generalized) linear model and briefly comment in Section 6 that it is sufficient to do this as it already gives assurance estimates of the mediation effects. This is problematic in two ways:
- It is a very strong assumption that ultimately determines what kind of $z$ you infer, for instance ensuring that you can never recover correctly a true $z$ for which the relationship is clearly non-linear. You may still be able to learn $z$ that is useful in estimating ACME, ADE and ATE, but I do not understand the value of the theoretical analysis of identifiability of the overall solution if you in the end make a strong assumption about the relationship that identifies the solution to a one that is defined by your assumption.
- You never present any empirical results with other choices, so we do not learn whether the claim of simple $g_{\gamma}$ being sufficient is correct or not. What if a non-linear mapping would clearly improve the estimates? If you want to claim a linear model is enough then you need to explicitly compare it against other choices. Now even the question of whether the model works at all with non-linear $g_{\gamma}$ remains open, and the reader starts to speculate that maybe the simplifying assumption was made because the solution was not identifiable in practice when a more flexible mapping was tried.

Related to the above, I am worried about validity of the claims. The paper is (at least implicitly) written in a way that it suggests the approach can recover the true effects with theoretical guarantees; the authors use statements like 'we prove that it is identifiable' based on theoretical analysis but do not discuss the relationship between the analysis and practical cases where all quantities are parameterised freely and estimated from the data. It remains highly uncertain what happens in practice and you empirically show that already for a very simple artificial data (Fig 3) you do not correctly identify the solution (the results show you capture the two modes nicely with the full model, but the result is definitely not identical to the true $p(z|t)$). There is definitely value in having a solid theoretical result even if it does not ensure guarantees in practice, but it would be important to discuss to which extent the guarantees hold in practice to avoid giving impression that the model really provides correct identifiable estimates. Section 6 now has a brief remark on this, reminding the reader that if $p(x|z)$ is incorrect then the estimates will be biased. A careful reader will know that this is always the case: You use here some neural network with simple likelihood and for any high-dimensional $x$ -- especially the kind of motivational examples you use like brain measurements -- this is obviously not going to be exact and in many cases also not invertible as required by the underlying theory (Eg. (9) or assumptions of Theorem 1). It would be better to make this explicit and state clearly the practical limitations also in Introduction and Conclusion, not just as side remark in Discussion.

The experimentation is in general okay but not self-contained. The method is compared against good baselines and the quality is measured with natural metrics. The ablation study is clear and useful (though I find it slightly confusing that $\alpha=-1$ refers to omitting one term -- maybe you could re-write Eq. (14) so that the multiplier is $\alpha-1$ and explain the reason for that already in Section 4.1?) bur the experiments in Sections 5.3 and 5.4 are cannot be properly understood by reading the paper. It is okay to leave the technical details in Appendix, but you are missing even the high-level description of how the data was simulated and the reader cannot understand the problem properly. For example, the read has no idea what 'concatenated MNIST' means and Tables 5-6 refer to $\nu$ that is not even mentioned in the paper. If you show results for two choices of that parameter you need to explain how the two cases differ qualitatively and what should we learn about the results.

---

> ### Author Response · Authors · 2024-01-10
> **Response to Reviewer 71Nt**
>
> Thank you so much for your constructive comments. Regarding your concerns under weaknesses and questions, we have listed our answers below.
>
> **The discussion on the effect of $g_{\gamma}$ being a simple model should be extended and the claims related to that need to be validated. You need to say how this choice relates to the theoretical analysis of identifiability of $z$ and you need to either remove the claim of simple $g_{\gamma}$ being enough or validate it empirically, by showing what happens if you use a more flexible network also in that part.**
>
> Thank you for pointing this out. Regarding your concerns, we try replacing the linear/logistic regression by a multi-layer perceptron for $g_{\gamma}$ and repeat the experiments in Sections 5.1 and 5.2. Tables 1, 2, and 4 have been updated to reflect the latest experimental results. In short, after using MLP for $g_{\gamma}$, we do observe further improved performance in case (b) with covariates. We have also included some relevant discussions in Section 6 and have removed the claims of simple $g_{\gamma}$ being enough. Please refer to the revised manuscript for more details.
>
> **The claims like "we prove that the true joint distribution over observed and latent variables is identifiable with the proposed method" should be re-phrased to make it clear that you have a theoretical proof of certain characteristics under a collection of assumptions (that are quite strong and will not hold in the kind of examples you use for motivating the work) but cannot prove that the proposed computational method would actually identify the correct latent variables.**
>
> Thank you for your invaluable suggestions. We have addressed the concerns raised by removing the specified claims in both Section 4.2 and Section 6, and we have added a sentence in the Limitation section to make it clear that the true distribution of $p(\boldsymbol{z}|t)$ is not guaranteed to be recovered, though we can still use the learned distribution to accurately estimate the direct and indirect effects. Also, we have given a brief discussion about the assumptions used to prove Theorem 1 at the end of Section 4.2, where we acknowledge that some of the assumptions are strong but were still used in some previous works.
>
> **The empirical experimentation needs to be made self-contained. Now many of the experimental details are only in Appendix and the reader cannot understand the experiments without consulting it, and many of the details needed for drawing conclusions are not provided at all.**
>
> Regarding your concern, we have added brief descriptions of the data generation process to Sections 5.1 – 5.3, including how we model the outcome $Y$, and how we construct the observed feature $X$ for the concatenated MNIST dataset. However, we refrain from transferring all the data preprocessing details in the Appendix to the main body as we are worried that such a move could potentially disrupt the overall flow of the experiment section.
>
> **Concatenation of $W$ and $T$ is technically correct, but likely to fail in practice when $W$ is very high-dimensional.**
>
> We have added one sentence in the Discussion section to discuss the dimensionality problem of $W$. We note the dimension of $W$ in the Jobs II dataset is around 20 which is already much higher than $T$, but our proposed method still demonstrates good performance. However, for $W$ with extremely high dimension, we recommend the readers to consider mapping it into a lower dimension before concatenation.
>
> **How were $\alpha$ and $\beta$ chosen for the full model in the experiments?**
>
> In the reported experiments, $\alpha$ and $\beta$ are manually tuned to ensure reasonable performance of the model.
>
> **Other minor comments**
>
> Figure 3 has been updated with a much higher resolution. Symbols such as $\alpha$, $\alpha_0$, $\beta$, and $\beta_0$ have been replaced by other characters to avoid overload. In Tables 3 and 4, our method (IMAVAE) has been moved to the last column.

---

> > ### Comment · Reviewer_71Nt · 2024-01-23
> > **Response**
> >
> > Apologies for the delay in responding to your clarifications.
> >
> > **Regarding your concerns, we try replacing the linear/logistic regression by a multi-layer perceptron for
> >  and repeat the experiments in Sections 5.1 and 5.2. Tables 1, 2, and 4 have been updated to reflect the latest experimental results**
> >
> > I appreciate this, but feel that in general the comparison is still very lightweight and does not provide a solid basis for making the claim of simple $g$ being sufficient in general; you only tried one other choice and observe it to have opposite effect in the two cases you considered, with the effect being fairly large in one of them. My conclusion from this comparison would be that the choice of $g$ remains a significant open challenge, and a practitioners interested in a well-performing model would need to separately start investigating the choice.
> >
> > **We have addressed the concerns raised by removing the specified claims in both Section 4.2 and Section 6, and we have added a sentence in the Limitation section to make it clear that the true distribution of is not guaranteed to be recovered, though we can still use the learned distribution to accurately estimate the direct and indirect effects.**
> >
> > The paper has improved in this respect, but I still find the paper to emphasize the identifiability too much compared to what the empirical results show. Already the name of IMAVAE as "Identifiable Mediation Analysis ..." promises too much, and I would prefer a naming convention that better reflects the properties you manage to confirm also empirically. Something like "Multivariate Mediation Analysis with Interpretable VAE (MMAIVAE)" would better capture the properties of the method, leaving the quantifier 'identifiable' to refer only to iVAE, not to the overall solution.
> >
> > **Regarding your concern, we have added brief descriptions of the data generation process to Sections 5.1 – 5.3, including how we model the outcome, and how we construct the observed feature for the concatenated MNIST dataset.**
> >
> > Thanks. Now that I understood how the concatenated MNIST was constructed, I must admit I do not understand at all why it was done that and what was the purpose. With the possible exception of the first digit, I would intuitively believe the individual digits of a cumulative distribution function to be effectively uniformly distributed and hence uninformative of the actual mediator. Could you clarify **why** the data was constructed like this and what is the actual goal? To me it looks like you now have an effectively univariate mediator that is mapped to a high-dimensional representation where some subset of the representation (the first digit of the concatenated image) directly maps mack to the actual mediator (assuming the model learns to solve MNIST image classification problem) and the remaining dimensions (the other three digits) are essentially random noise that is largely uninformative of the true mediator. This is maybe not how you think about it, but nevertheless I would expect a proper explanation.
> >
> >
> > Overall, I feel the paper has improved but the changes to not properly address the main concerns and I see no reason to change my overall evaluation at this point.
> >
> > Minor comment: The updated version seems to be missing the abstract completely.

---

> > > ### Author Response · Authors · 2024-01-26
> > > **Response to feedback from Reviewer 71Nt**
> > >
> > > Thank you for your further feedback on our previous response. Regarding your question on the MNIST data generation, we agree that such a strategy to map the mediator into a high-dimensional representation may not be conventional, but as we stated in Section 5.3, we adopt this approach to closely resemble the methodology outlined in simulation 1 of Nath et al. [1] to facilitate a fair comparison with benchmark models, i.e., Deep LSEM, Shallow LSEM, and SVR in Table 5. According to Section 2.2 in Nath et al., the goal of this transformation between $norm.cdf(Z)$ and the MNIST digits is to simulate a situation in which the latent mediator $Z$ is a complex and nonlinear function of an observed set of mediator variables.
> > >
> > > [1]. Nath, T., Caffo, B., Wager, T., & Lindquist, M. A. (2023). A machine learning based approach towards high-dimensional mediation analysis. NeuroImage, 268, 119843.

---

### Review · Reviewer_7yag · 2023-12-28

**Summary Of Contributions:**

This work aims to extend mediation analysis by proposing a method for estimating direct and indirect effects when the mediator is an unobserved and real-valued variable. The recovery of direct and indirect effects with unobserved mediators builds upon previous work in causal inference and identifiable latent variable models.

**Audience:**

No

**Broader Impact Concerns:**

I see no immediate ethical concerns in the context of this work.

**Claims And Evidence:**

No

**Requested Changes:**

* It is not completely clear whether the results in sec. 3.1 are for categorical $\boldsymbol{x}, \boldsymbol{z}$ variables, whereas the rest of the paper focuses on real-valued variables. Could you please clarify this? Also, where exactly are the derivations in sec. 3.1 used afterwards?
* At the end of sec. 3.1, the authors write: _"the total effect is just a combination of the direct and indirect effects"_. Please elaborate on this: In linear models, an additive decomposition of the total causal effect into direct and indirect contributions exists, but this generalization is not valid in non-parametric models ([1], Sec. 3.5), due to interactions between treatment and mediator, known as moderation (see [2]).
* _"Our primary objective is to learn a disentangled representation of the true mediator in the latent space so that the statistical distance between $p(\boldsymbol{z} \mid t=0)$ and $p(\boldsymbol{z} \mid t=1)$ can be better estimated."_ I do not understand what the authors meant here: please explain.
* Thm. 1 in (Khemakhem et al., 2020) requires a sufficient number of distinct values for the conditioning variable. Given the binary treatment, does this assumption pose a challenge for the application of Thm. 1 to Case (a)?
* The way objective (14) is currently written is confusing. IIt would be more clear if the authors could use different symbols, or different superscripts/subscripts to denote different losses.
* It is also not fully clear how the different terms in the loss enter the proof of Thm. 1 in Appendix A, which is based on the proof of Thm. 1 of (Khemakhem et al., 2020). Could you please explain this?
* After the application of Step 1 in the proof of Thm. 1 in Appendix A, the variable $\boldsymbol{z}$ is essentially observed (albeit not from the point of view of nonlinear ICA/disentanglement, since it would still be entangled). This is because the function is injective and information about the latent variable is completely preserved in the observed variable when the additive noise is removed. I'm wondering whether removing the additive noise would not be sufficient for the author's objectives---in which case, it would be unclear why the iVAE framework should be used. Could you please explain why disentanglement is further needed?

**References:**

[1] Pearl, Judea. "Direct and indirect effects." Probabilistic and causal inference: the works of Judea Pearl. 2022. 373-392.

[2] Pearl, Judea. "Interpretation and identification of causal mediation." Psychological methods 19.4 (2014): 459.

**Strengths And Weaknesses:**

**Strengths:**

The idea of integrating classic results in causal inference with more modern results on identifiable latent variable models is commendable.

**Weakness:**

Unfortunately, there are several issues with the clarity of the paper.

To my understanding, the authors need to identify certain conditional probabilities, which would allow them to evaluate the quantities described in equations (1)-(3). The reason for employing the iVAE framework, designed primarily for the separation or disentanglement of conditionally independent latent variables, remains unclear. Reconstructing (conditional) probabilities of the latent variables should not require disentanglement or source separation, since these would be preserved by any measure-preserving automorphisms [1] (which would, in turn, yield entangled latent variables). Moreover, it is unclear why the assumption of conditional independence of the latent variables $\boldsymbol{z}$ is relevant for this work, whereas it is of central importance in the nonlinear ICA work by (Khemakhem et al., 2020).

In short, it is not clear why conditional independence is relevant, and why disentanglement is desirable.

Finally, I do not see how Assumption 4 in Thm. 1, which requires a sufficient number of distinct values for the conditioning variable, can be meaningfully satisfied by binary treatment variables in Case (a), as described in section 4.1 of the manuscript.

Due to the aforementioned points, I find it difficult to assess the clarity and interest of the paper's claims and findings based on the current status of the manuscript. Please find a more detailed set of questions and comments in the "Requested Changes" section.

Minor: I also found it difficult to understand what are the original theoretical contributions presented in this work. The results in section 3.1 may be entirely taken from previous work (Pearl, 2012), although this is not fully clear from the paper. The proof of Theorem 1 appears to be the same as Thm. 1 of (Khemakhem et al., 2020). As I mentioned, the combination of existing results may, in principle, be of interest in itself. Nonetheless, I would appreciate it if the authors could be more explicit about it.

**References:**

[1] Xi, Quanhan, and Benjamin Bloem-Reddy. "Indeterminacy in generative models: Characterization and strong identifiability." International Conference on Artificial Intelligence and Statistics. PMLR, 2023.

---

> ### Author Response · Authors · 2024-01-10
> **Response to Reviewer 7yag (1/2)**
>
> Thank you so much for your constructive comments. Regarding your concerns under weaknesses and questions, we have listed our answers below.
>
> **It is not completely clear whether the results in sec. 3.1 are for categorical $\boldsymbol{x}$, $\boldsymbol{z}$ variables, whereas the rest of the paper focuses on real-valued variables. Could you please clarify this? Also, where exactly are the derivations in sec. 3.1 used afterwards?**
>
> We present the derivation in Section 3.1 in the context of discrete variables just to be consistent with the one shown in Pearl (2012) [1]. However, this derivation can be naturally extended to the case of continuous variables by replacing the summations with integrals. We have included an additional sentence in Section 3.1 to clarify this point. As we claim at the beginning of Section 3.1, the derivations mainly state that the direct, indirect, and total effects are recoverable even with unobserved mediator $Z$. Failing to explicitly derive this point might lead to a perceived conflict between the sequential ignorability assumption (Assumption 2 in Section 3) and the unobserved nature of $Z$.
>
> Regarding the theoretical contributions, we would like to argue that our derivations in this section is different from the one presented by Pearl (2012) [1] as Pearl followed a confounding structure while we follow a mediation structure, i.e., $Z$ acts as a mediator instead of a confounder.
>
> [1]. Pearl, J. (2012). On measurement bias in causal inference. arXiv preprint arXiv:1203.3504.
>
> **At the end of sec. 3.1, the authors write: "the total effect is just a combination of the direct and indirect effects". Please elaborate on this: In linear models, an additive decomposition of the total causal effect into direct and indirect contributions exists, but this generalization is not valid in non-parametric models due to interactions between treatment and mediator, known as moderation.**
>
> Thank you for bringing this to our attention. However, we are not aware of any claims relevant to nonparametric models in Section 3.5 of [1]. Under Pearl’s definition, the natural direct effect (NDE) corresponds to our definition of $ADE(t = 0)$ and the natural indirect effect (NIE) corresponds to our definition of $ACME(t = 0)$. In Section 3.5 of [1], although Equation 24 only holds for linear models, we believe Equations 22 and 23 hold in general, which also corresponds to Equation 12 in [2]. Consequently, we maintain the accuracy of our assertion at the end of Section 3.1, which states that the direct effect can be recovered if we are able to recover the indirect effect and the total effect.
>
> [1]. Pearl, J. (2022). Direct and indirect effects. In Probabilistic and causal inference: the works of Judea Pearl (pp. 373-392).
>
> [2]. Pearl, J. (2014). Interpretation and identification of causal mediation. Psychological methods, 19(4), 459.
>
> **“Our primary objective is to learn a disentangled representation of the true mediator in the latent space so that the statistical distance between $p(\boldsymbol{z}|t = 0)$ and $p(\boldsymbol{z}|t = 1)$ can be better estimated." I do not understand what the authors meant here: please explain.**
>
> With this argument, we want to convey to the readers that we believe disentanglement of the conditional prior, i.e., $p(\boldsymbol{z}|t = 0)$ and $p(\boldsymbol{z}|t = 1)$, can be helpful for the pure technical goal of estimating mediation effects. However, we are not stating any theoretical guarantees here. To avoid confusion, we have rephrased this sentence and have included appropriate citations.
>
> To see how disentangled latent representations helps with mediation effect estimation, we have added another ablation study in Section 5.1 where we replace the iVAE backbone in our framework with a conventional VAE (i.e., $\boldsymbol{z}$ is no longer dependent on $t$ in the latent space, leading to entangled representations).  This structure does not capture the treatment effect on the latent space, causing  We then observe that both direct and indirect effect estimations to become much worse.

---

> > ### Comment · Reviewer_7yag · 2024-01-17
> > **Polite disagreement with assertion**
> >
> > "the total effect is just a combination of the direct and indirect effects"
> >
> > I respectfully disagree with the assertion, as I find it to be inaccurate or potentially misleading, depending on the meaning you assign to certain terms. Please see, e.g., [1], section IV.G:
> >
> > > _Can the total causal effect be decomposed into a sum of direct and indirect contributions?_ While such an additive decomposition indeed exists for linear models, it does not hold in general due to possible interactions between treatment and mediator, referred to as _moderation_. Direct and indirect effects are not uniquely defined in general, but depend on the value of the mediator. Counterfactual quantities such as NDE and NIE are thus useful tools to measure some average form of direct and indirect effect with a meaningful interpretation.
> >
> > [1] von Kügelgen, Julius, Luigi Gresele, and Bernhard Schölkopf. "Simpson's paradox in Covid-19 case fatality rates: a mediation analysis of age-related causal effects." IEEE Transactions on Artificial Intelligence 2.1 (2021): 18-27.

---

> ### Author Response · Authors · 2024-01-10
> **Response to Reviewer 7yag (2/2)**
>
> **Thm. 1 in (Khemakhem et al., 2020) requires a sufficient number of distinct values for the conditioning variable. Given the binary treatment, does this assumption pose a challenge for the application of Thm. 1 to Case (a)?**
>
> We believe that the presence of a binary treatment will not pose a challenge to the application of Theorem 1 in Case (a), as evidenced by Khemakhem et al.'s analogous applications in their work. For example, in Section 5.1 and Figure 1 of [1], the conditioning variable $\boldsymbol{u}$ serves as the segment label, taking only five distinct values. Moreover, it is crucial to note that the quantities $n$ and $k$ in Assumption 4 of Theorem 1 correspond to the dimension of $\boldsymbol{z}$ and the dimension of the sufficient statistic, respectively. Importantly, neither of these quantities is directly contingent on the number of distinct values assumed by $\boldsymbol{u}$. In Section B.2.3 of the Appendix of [1], Khemakhem et al. provide a simple example where $n = 2$ and $k = 1$, demonstrating the satisfaction of Assumption 4 without any direct dependence on the number of distinct values of $\boldsymbol{u}$.
>
> [1]. Khemakhem, I., Kingma, D., Monti, R., & Hyvarinen, A. (2020, June). Variational autoencoders and nonlinear ica: A unifying framework. In International Conference on Artificial Intelligence and Statistics (pp. 2207-2217). PMLR.
>
> **The way objective (14) currently written is confusing. It would be clearer if the authors could use different symbols, or different superscripts/subscripts to denote different losses.**
>
> Thank you for your suggestion. We have added different superscripts to each individual term in Equation (14) to make it clearer.
>
> **It is also not fully clear how the different terms in the loss enter the proof of Thm. 1 in Appendix A, which is based on the proof of Thm. 1 of (Khemakhem et al., 2020). Could you please explain this?**
>
> Similar to Khemakhem et al., Theorem 1 provides a basic form of identifiability for the generative model $p_{\theta}(\boldsymbol{x}, \boldsymbol{z}, y|\boldsymbol{u})$, which is mainly associated with $p_{\theta}(\boldsymbol{x}, \boldsymbol{z}|\boldsymbol{u})$ in the ELBO term and $p_{\textbf{h}}(y|\boldsymbol{z}, \boldsymbol{u})$ in the prediction term in Equation 14. The reconstruction term is explicitly written out in Equation 14 just to emphasize that the loss consists of both reconstruction and KL divergence. It worths noting that for the predictor $p_{\textbf{h}}(y|\boldsymbol{z}, \boldsymbol{u})$, we only claim the identifiability of $\boldsymbol{\psi}$’s projection onto $\mathcal{Z}$ but not onto $\mathcal{U}$.
>
> **I'm wondering whether removing the additive noise would not be sufficient for the author's objectives---in which case, it would be unclear why the iVAE framework should be used. Could you please explain why disentanglement is further needed?**
>
> Please refer to our answer to your previous question on the disentanglement of latent representations.

---

> > ### Comment · Reviewer_7yag · 2024-01-17
> > **Potential misunderstanding of the theory in Khemkahem et al.**
> >
> > > For example, in Section 5.1 and Figure 1 of [1], the conditioning variable serves as the segment label, taking only five distinct values.
> >
> > Indeed, the authors require $5$ distinct values for $n=2$, and $k$ at most equal to $2$. This is fine, because the values required in their assumption are at least $nk+1=4+1=5$. The authors never use $2$ distinct values only---see below.
> >
> > > In Section B.2.3 of the Appendix of [1], Khemakhem et al. provide a simple example [...] demonstrating the satisfaction of Assumption 4 without any direct dependence on the number of distinct values of $\boldsymbol{u}$.
> >
> > Respectfully, this is wrong and based on a misunderstanding of section B.2.3 of Khemakhem et al. In fact, in the example of section B.2.3, the authors require that the auxiliary variable takes three distinct values, $u_0=1$ and $u_1, u_2$ _"distinct non-zero scalars"_. That is: for $n=2$ and $k=1$, the authors require at least $3=nk+1$ distinct values $u_0, u_1, u_2$, as by their Assumption _(iv)_. This is necessary for their proof to work.
> >
> > With a binary treatment, only two values of $\boldsymbol{u}$ are given. So even if it were the case that $k=1$, there's no solution to $2=nk + 1$ other than $n=1$, i.e., a unidimensional latent variable. So binary treatment is problematic, in the sense that it is not possible that, with a binary treatment, Assumption _(iv)_ holds for a latent dimensionality which is higher than one---in which case it is pointless to talk about disentanglement.

---

> ### Comment · Reviewer_7yag · 2024-01-17
> **Disentanglement and the iVAE**
>
> Thank you for your response regarding disentangled representations. Based on your response and the modifications in the paper, my understanding is that disentangled representations appear to be heuristically helpful for estimating mediation effects, whereas you do not make any theoretical arguments about it. I still find it somewhat perplexing that the iVAE, whose primary motivation is to discover identifiable representations, plays a significant role in this work.

---

> ### Author Response · Authors · 2024-01-19
> **Response to follow-up questions from Reviewer 7yag**
>
> Thank you so much for your feedback on our response. Regarding your follow-up questions, please see our answers below. The corresponding modifications are highlighted in blue in the updated manuscript.
>
> **“The total effect is just a combination of the direct and indirect effects" – I respectfully disagree with the assertion, as I find it to be inaccurate or potentially misleading, depending on the meaning you assign to certain terms.**
>
> We apologize for the confusion. We note that we have implicitly assumed no interaction between $Z$ and $T$ in our model formulation as we do not include any interaction terms between $Z$ and $T$ when predicting $Y$. We have modified the sentence describing the relationship between the direct, indirect, and total effects at the end of Section 3.1 to avoid confusion, and have added this point to the limitation section.
>
> **In Section B.2.3 of the Appendix of [1], Khemakhem et al. provide a simple example [...] demonstrating the satisfaction of Assumption 4 without any direct dependence on the number of distinct values of $\boldsymbol{u}$. – Respectfully, this is wrong and based on a misunderstanding of section B.2.3 of Khemakhem et al.**
>
> We appreciate your insightful observation. We agree that Assumption 4 of Theorem 1 in the manuscript is generally applicable to case (b). However, for case (a), it holds only when $d = k = 1$, signifying a scenario where $\boldsymbol{z}$ is one-dimensional, and the number of sufficient statistics is 1. To address this clarification, we have added a corresponding sentence at the end of Section 4.2 in the manuscript.

---

### Author Response · Authors · 2024-01-10
**Manuscript has been updated**

Dear reviewers,

Thank you for your time and effort in reviewing our paper. We have uploaded an updated version of our manuscript. For your convenience, we have highlighted all changes in blue and we will summarize the major changes below:

• We modify the claim about the disentangled representations of the conditional prior and include relevant citations at the beginning of Section 4.

• For Sections 5.1 and 5.2, we add another variant of our proposed framework with $g_{\boldsymbol{\gamma}}$ implemented as a multi-layer perceptron (MLP) and update the tables and texts accordingly.

• We add an ablation study (Section 5.1.1) to **empirically** illustrate how disentanglement in $p(\boldsymbol{z}|t)$ helps accurately estimate the mediation effect.

• We modify the Discussion section (Section 6) to elaborate in more detail about the inference procedure of direct and indirect effects and to discuss the limitations of assumptions and scope of the identifiability theory presented in this paper.

Please feel free to post more comments or questions in case our answers do not fully address your concerns.

Best regards,

Authors of Submission 1910

---

### Decision · Action_Editor_uQu8 · 2024-01-30

**Recommendation:** Reject

**Comment:**

Though this paper has some worthwhile components to it, the reviewers were fairly unanimous that the empirical evidence provided does not fully justify the claim that this technique could be used on real-world data for identifying multi-variate, indirectly-observed mediators. With the added concern that the main theorem has potentially limited applicability, that renders this paper's claims not well supported. As such, a decision of 'reject' was natural. If the authors can provide data that addresses some of these concerns more robustly (e.g. more robust demonstration that the form of $g_\gamma$ can be dealt with appropriately for real-world applications), then a re-submission would be considered.

**Audience:**

This is very appropriate for the TMLR audience.

**Claims And Evidence:**

In this paper the authors present a VAE-based technique for causal mediation analysis designed to handle multi-dimensional, indirectly-observable mediators. The authors show with both synthetic data and semi-synthetic data (ephys, concatenated MNIST, Jobs II) that their method can reduce the error in identifying the effect. The authors claim to prove that the true joint distribution over observed and latent variables is identifiable with their method, and argue that their empirical data supports the claim that this method is an advance for doing causal mediation analysis in real-world applications.

The reviewers were concerned that this paper does not sufficiently support its claims. First, there was broad agreement amongst the reviewers that though there is a fair bit of data, the empirical evidence provided does not clearly demonstrate the advantages of the proposed method, nor strongly support the theoretical claims of the paper. The authors attempted to assuage these concerns with some new experiments, but the reviewers felt that these experiments did not actually support the claims robustly, either. There was also a concern that the main theorem may actually not be applicable to many real-world cases (those with binary treatment variables).

**Resubmission Of Major Revision:**

The authors may consider submitting a major revision at a later time.